# Balancing the Conservation and Poverty Eradication: Differences in the Spatial Distribution Characteristics of Protected Areas between Poor and Non-Poor Counties in China

**Luqiong Fan** [1,2,3]**, Chunting Feng** [2,3,]*****, Zhixue Wang** [2,3]**, Jing Tian** [2,3]**, Wenjie Huang** [4] **and Wei Wang** [2,3,]***** 

1   School of Ecology and Environment, Zhengzhou University, Zhengzhou 450001, China; flq0503@163.com
2   State Key Laboratory of Environmental Criteria and Risk Assessment, Chinese Research Academy of Environmental Sciences, 8 Anwai Dayangfang, Chaoyang District, Beijing 100012, China; 18434375143@163.com (Z.W.); tianjing970114@163.com (J.T.)
3   Biodiversity Research Center, Chinese Research Academy of Environmental Sciences, Beijing 100012, China
4   Teachers College, Beijing Union University, Beijing 100011, China; hwj5242@163.com
*   Correspondence: fengchunting212@163.com (C.F.); wang.wei@craes.org.cn (W.W.)

**Abstract:** Understanding the spatial distribution characteristics of protected areas is the basis to balance the conservation and regional development. With the increasing number and area of protected areas, China has also made decisive progress in the fight against poverty. However, the spatial distribution characteristics of various types of protected areas in poor counties in China are still unclear and lacking further analysis on the differences compared to non-poor counties. Here, we first integrated the spatial distribution data of 8133 protected areas in China and overlaid them with 832 poor counties. Then we explored the spatial distribution characteristics of protected areas and the relationship with socio-economic and natural environment in poor and non-poor counties. The results showed that the area coverage of nature reserves in poor counties in China was significantly higher than that in non-poor counties ($p < 0.001$), while the area coverage of natural parks in non-poor counties was significantly higher than that in poor counties ($p < 0.05$). The area coverages of protected areas in poor counties in Northeast ($p < 0.05$), Southwest ($p < 0.001$), Central ($p < 0.05$), and East China ($p < 0.01$) were significantly higher than that in non-poor counties. Furthermore, the area coverage of nature reserves in poor counties was significantly positively correlated with mean elevation ($p < 0.001$), and the area coverage of natural parks in non-poor counties was significantly positively correlated with road network density ($p < 0.05$) and negatively correlated with the proportion of farmland ($p < 0.001$). This study can provide a reference to help China and other similar countries in the establishment of protected area systems to balance the conservation and poverty eradication for regional sustainable development.

**Keywords:** protected areas; poor counties; biodiversity; socio-economic; spatial distribution

## 1. Introduction

Protected areas are the most direct and effective initiative to conserve biodiversity [1–3], and the services provided by good ecosystems in protected areas are of great value to human livelihoods, health and well-being [4,5]. Generally, human activities interfere with protected areas may cause a sharp decline in biodiversity if socio-economic development is pursued blindly beyond the protected areas' carrying capacity. At the same time, protected areas may have positive or negative effects on local socioeconomic development [6–8]. Several studies analyzed the impact of protected areas on local development by using quasi-experimental matching methods [9] and constructing poverty indices [7,10], and a large number of cases showed that protected areas bring positive effects on poverty eradication (Such as PANORAMA case platform, https://panorama.solutions/en, accessed on 20 November 2021) [4]. While some reports found that protected areas had nonsignificant positive

effects on regional socioeconomic development because they limited the direct use of protected areas by local community residents [11–13], or even had negative effects on local development [14,15]. Therefore, the relationship between protected areas and regional socio-economic development is very close and has become one of the hot issues in the field of protected areas.

Understanding the spatial distribution characteristics of protected areas is the basis to balance the conservation and regional development. Recent studies had focused on the spatial distribution pattern of protected areas in relation to the environment and the degree of land development. It had been found that protected areas were mostly located in remote areas with higher altitudes and lower agricultural value [16]. For example, larger protected areas in the United States were mostly located in Alaska and western regions with lower agricultural value, while protected areas in the eastern regions with higher agricultural value were generally smaller in size [17]. In addition, most strict protected areas (i.e., IUCN categories I and II) were distributed in higher and steeper lands away from roads and urban centers than less intensively managed reserves (i.e., IUCN category III-VI) [16]. But the less intensively managed reserves can allow people to generate revenue through tourism or harvesting and transporting plants and animals for sale in markets [8], which could obtain more opportunity to develop and face higher intensity of human disturbance [18] and greater deforestation pressure [19] at the same time. Considering that poor areas and non-poor areas differ in terms of environmental and land development levels, it is essential to further explore whether there are differences in the spatial distribution of different types of protected areas in poor and non-poor areas, with the aim to support policy formulation and implementation processes to balance the conservation and poverty eradication for regional sustainable development.

China is one of the mega-biodiversity countries in the world. Since the establishment of the first nature reserve in 1956, China has built about 11,800 protected areas with an area of over 170 million hectares by the end of 2020, accounting for about 18% of the national land area, which has achieved the Aichi targets for the proportion of land area for protected areas [20]. With the increasing number and area of protected areas, China has also made decisive progress in the fight against poverty [21]. China achieved full poverty eradication by the end of 2020 (The National Rural Revitalization Administration, 2020), which has achieved the poverty reduction targets of "*Transforming our World: The UN 2030 Agenda for Sustainable Development*" 10 years ahead of schedule [22]. Recent studies showed that poor regions in China were strongly coupled with ecologically fragile areas in terms of geographical distribution [23–25], which also had high biodiversity [26–28]. For example, national nature reserves [29], national key ecological function areas [30,31], and priority areas for biodiversity conservation [32] are mainly located in areas with low population density and more backward development. Meanwhile, these regions were also mostly semi-enclosed with poor infrastructure and limited developing opportunities. However, the spatial distribution characteristics of various types of protected areas in poor counties in China are still unclear and lacking further analysis on the differences compared to non-poor counties.

Thus, we explored the differences in the spatial distribution characteristics of protected areas between poor and non-poor counties in China. We integrated the spatial distribution of 8133 protected areas in China and overlaid them with the 832 poor counties that have completed poverty eradication from 2015–2020 as well as the other 2023 non-poor counties. We then explored the relationship between the area coverage of different types of protected areas and the socio-economic and natural environment in poor and non-poor counties, with the aim to provide a reference to help China and other similar countries in the establishment of protected area systems to balance the conservation and poverty eradication.

## 2. Materials and Methods

### 2.1. Data

In recent years, in order to strengthen biodiversity conservation, China is accelerating the construction of a protected areas system with national parks as the mainstay. In June 2019, the General Office of the Communist Party of China and the State Council General Office issued "*A Guideline on Establishing a System of Protected Natural Areas with National Parks as its Mainstay*", with the aim to reclassify the protected areas to 3 types, national parks, nature reserves, and natural parks. In October 2021, China officially established the first batch of national parks, including Sanjiangyuan, Giant Panda, Northeast China Tiger and Leopard, Hainan Tropical Rainforest, and Wuyi Mountain.

Since national parks have not been officially established in China as of the end of 2020, we did not include national parks as a type of protected areas in this study. The types of protected areas in China involved in this study were nature reserves and natural parks up to the end of 2020. The original distribution data of protected areas were obtained from the State Ministry of Ecology and Environment, Provincial Department of Ecology and Environment, Provincial Department of Forestry and Grassland, and management bureaus of protected areas. We collected and established the distribution geodatabase of 8133 protected areas, including 2673 nature reserves and 5460 natural parks (e.g., forest parks, wetland parks, geological parks, desert parks, marine parks, and mine parks, excluding scenic spots and aquatic germplasm resource reserves) (Table 1).

**Table 1.** Data type and sources.

| Data Type | Data Sources |
| --- | --- |
| Protected areas | State Ministry of Ecology and Environment, Provincial Department of Ecology and Environment, Provincial Department of Forestry and Grassland, and management bureaus of protected areas |
| Poor counties and Non-poor counties | The National Rural Revitalization Administration (http://www.nrra.gov.cn/, accessed on 1 August 2021) Ministry of Natural Resources of the People's Republic of China (http://www.mnr.gov.cn/, accessed on 1 August 2021) |
| Road network | 2015 national road data (30 m spatial resolution) |
| Population | National Bureau of Statistics, 2016 (http://www.stats.gov.cn/, accessed on 31 October 2021) |
| GDP | National Bureau of Statistics, 2016 (http://www.stats.gov.cn/, accessed on 31 October 2021) |
| Farmland | 2015 national land use data (30 m spatial resolution) |
| Elevation | 30 m ASTER GDEM V2 from Geospatial Data Cloud (http://www.gscloud.cn/, accessed on 5 November 2021) |

The former Aid-the-Poor Development Office of the State Council (currently the National Rural Revitalization Administration) identified 832 poor counties in 2012 (http://www.nrra.gov.cn/, accessed on 1 August 2021) used the indicators highly correlated with the degree of poverty, such as per capita county GDP, per capita county general budget revenue and per capita net income of county farmers. We collected the spatial distribution and area of the 832 poor counties that have completed poverty eradication from 2015–2020, as well as 2023 non-poor counties.

Relevant socioeconomic and environmental data include road network density, population density, GDP (Gross Domestic Product) per capita, farmland, and elevation data. The total population and GDP data were obtained from the "*China County Statistical Yearbook 2015*" (National Bureau of Statistics, 2016). The road network and farmland were obtained from the 2015 national road data and land use data (30 m spatial resolution), respectively. And the elevation data were obtained from the 30 m ASTER GDEM V2 (Table 1).

The area of protected areas in each county was obtained by overlaying county-level administrative division maps and distribution maps of protected areas by GIS software ArcGIS 10.6. County population density and GDP per capita were calculated through the statistics of China County Statistical Yearbook, and road network density, mean elevation, and farmland ratio were processed using ArcGIS 10.6 overlay analysis.

Since China's poverty alleviation policies used to cover only the mainland, all data in this article did not include Hong Kong, Macao, and Taiwan.

*2.2. Methods*

The research methodology of this study is shown in Figure 1. First, we analyzed the spatial distribution of protected areas in poor counties and non-poor counties, respectively. Then, we divided China's administrative regions into seven sub-regions: Northeast, Northwest, Southwest, North, South, Central, and East, and explored the spatial distribution of protected areas in poor and non-poor counties in different sub-regions. Last, we tested the relationship between spatial distribution of different types of protected areas and related socio-economic and environmental factors.

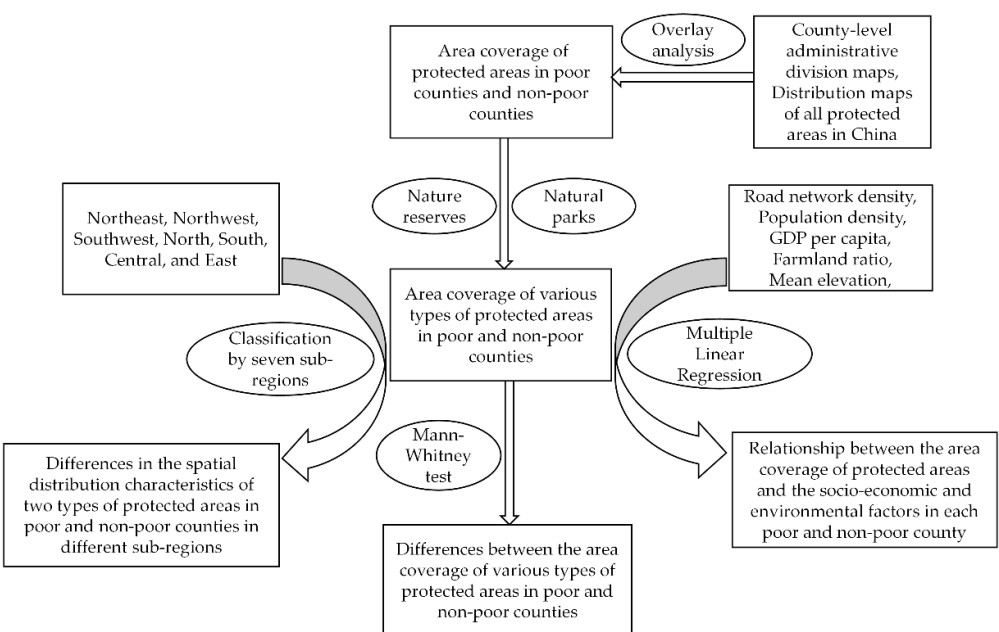

**Figure 1.** Methodology flowchart.

2.2.1. Analysis of the Overall Spatial Distribution of Protected Areas

We overlaid the county-level administrative division maps with the distribution maps of all protected areas in China, and calculated the area coverage of protected areas in poor counties and non-poor counties, respectively (Figure 2a, Table S1). According to the "*A Guideline on Establishing a System of Protected Natural Areas with National Parks as its Mainstay*", the primary function of nature reserves is typical ecosystem protection, species and habitat protection, natural relic protection, and natural science research. In addition, the "*Regulations of the People's Republic of China on Nature Reserves*" also clearly stipulate that logging, grazing, hunting, fishing, medicine collection, reclamation, burning, mining, quarrying, and sand digging are prohibited in nature reserves. Therefore, nature reserves generally have strict management intensity which may limit regional economic development. By contrast, the function of natural parks includes landscape protection, recreation, resource availability or resource production, balance nature conservation and resource utilization in the intensity of management and conservation. Natural parks may obtain more opportunity to promote economic development by developing tourism and providing resources. Thus, we divided the protected areas into two types, nature reserves and natural parks, and calculated the area coverage of them in poor counties and non-poor

counties, respectively (Figure 2b,c and Table S1). The differences between the area coverage of protected areas in poor and non-poor counties were compared by Mann-Whitney test.

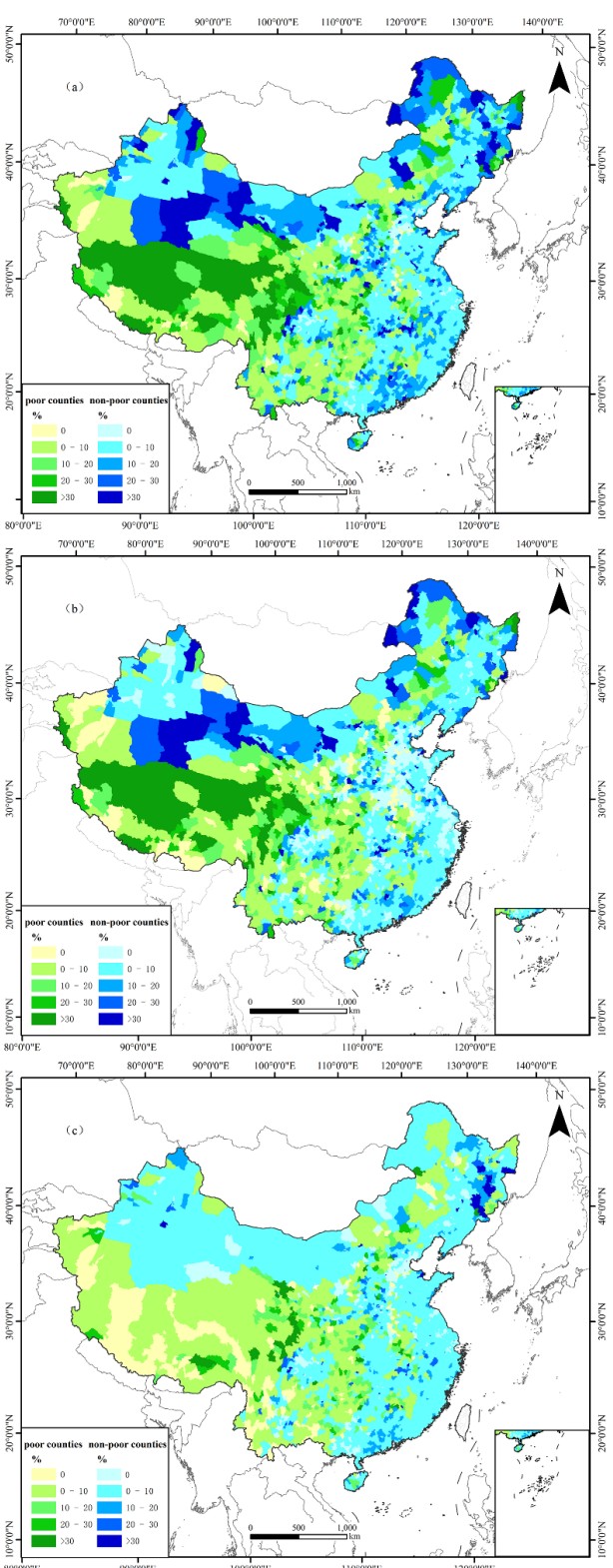

**Figure 2.** Area coverage of protected areas (**a**), nature reserves (**b**), and natural parks (**c**) in each county.

### 2.2.2. Spatial Distribution of Protected Areas in Different Sub-Regions

We divided China's administrative regions into seven sub-regions: Northeast, Northwest, Southwest, North, South, Central, and East. To investigate the inter-regional dif-

ferences in the spatial distribution of protected areas in poor and non-poor counties, we analyzed the spatial distribution characteristics of protected areas in poor and non-poor counties in different sub-regions. We further analyzed the differences in the spatial distribution characteristics of two types of protected areas, nature reserves and nature parks, in poor and non-poor counties in different subregions, respectively. The differences between the area coverage of protected areas in poor and non-poor counties in different subregions were compared by Mann-Whitney test respectively.

### 2.2.3. Relationship with Socio-Economic and Environmental Factors

To further explore the potential reason that may influence the distribution of protected areas, we analyzed the relationship between the area coverage of protected areas and the socio-economic and environmental factors in each poor and non-poor county. According to recent reports [16,29], A number of studies have shown that protected areas are usually biased toward sites with lower productivity and economic value, higher elevation, and farther from road cities [16,29,33]. In addition, the degree of human disturbance is widely used as one of the indicators of threats or pressures on natural ecosystems [34]. Thus, we selected five socio-economic and environmental factors, including the road network density, population density, per capita GDP, farmland ratio, and mean elevation. The population density and GDP per capita of each county were obtained from the statistical calculation of "*China County Statistical Yearbook*", and the road network density, farmland ratio, and mean elevation were calculated by spatial analyses.

All the data were standardized with Stata 16 to remove the magnitudes and were calculated in poor and non-poor counties, respectively. The area coverage of protected areas in each county was set as the dependent variable and the 5 socioeconomic and environmental factors of each county were set as the independent variables. We used the multiple linear regression to explore the relationship between the area coverage of protected areas and the socio-economic and environmental factors. The formula of multiple linear regression is as follows (1).

$$Y_i = \beta_0 + \beta_1 X_1 + \beta_2 X_2 + \beta_3 X_3 + \beta_4 X_4 + \beta_5 X_5 + \mu \tag{1}$$

where $Y_i$ is the area coverage of protected areas in the county; $\beta_0$ is the regression constant; $\beta_1, \ldots, \beta_5$ are the regression coefficients; $X_1, \ldots, X_5$ denote road network density, mean elevation, population density, GDP per capita, and farmland ratio, respectively; $\mu$ is the random error.

## 3. Results

### 3.1. Overall Spatial Distribution of Protected Areas in Poor and Non-Poor Counties

Among the 832 poor counties in China, there were 787 poor counties with protected areas distributed, in which 77 counties had more than 30% coverage of protected areas, accounting for 9.25% of the total number of poor counties (Figure 2a). In contrast, the number of non-poor counties with more than 30% coverage of protected areas was 111, accounting for 3.86% of the total number of non-poor counties. The area coverage of protected areas in poor counties was significantly higher than that in non-poor counties ($p < 0.001$) (Figure 3a).

In terms of different types of protected areas, the area coverage of nature reserves in poor counties in China was significantly higher than that in non-poor counties ($p < 0.001$) (Figures 2b and 3b). While the area coverage of natural parks in non-poor counties was significantly higher than that in poor counties ($p < 0.05$) (Figures 2c and 3c).

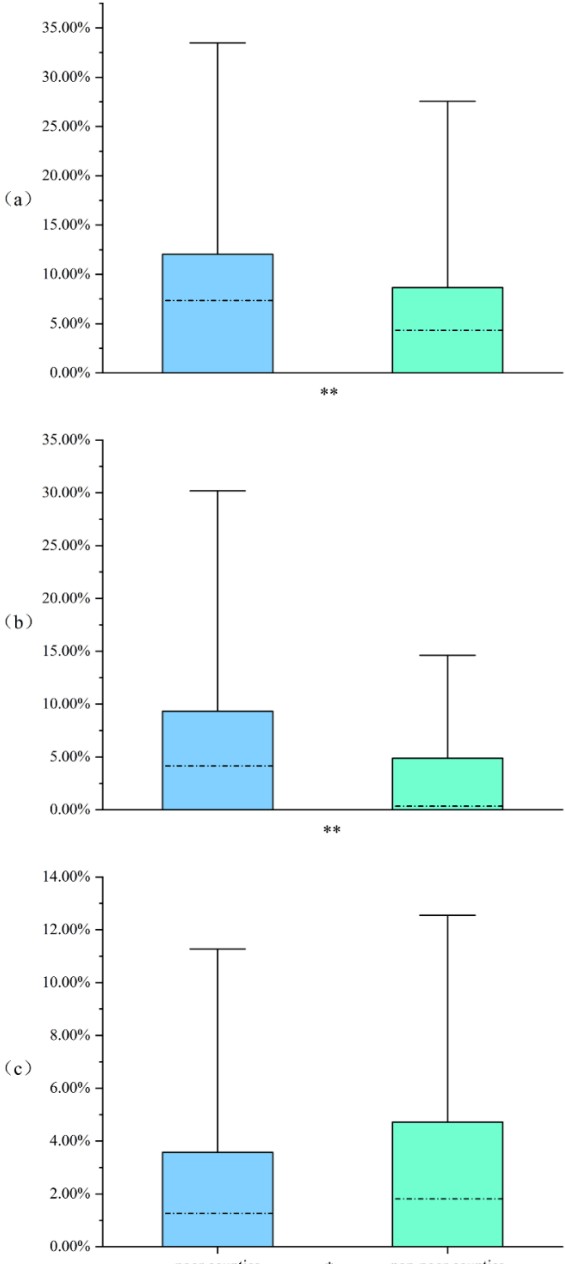

**Figure 3.** Significant differences in area coverage of protected areas (**a**), nature reserves (**b**), and natural parks (**c**) between poor and non-poor counties (* $p < 0.05$, ** $p < 0.001$).

### 3.2. Spatial Distribution of Protected Areas in Poor and Non-Poor Counties in Different Sub-Regions

In general, the area coverage of protected areas in poor counties showed a trend of decreasing from the west regions to the east regions. The area coverages of protected areas in poor counties were relative higher in Northeast, Northwest, and Southwest China. While the East China had the relative lower coverage of protected areas. In contrast, the coverage of protected areas in non-poor counties was higher in the north regions than in the south regions. The area coverages of protected areas in non-poor counties were relative higher in Northeast and Northwest China, and relatively lower in Southwest and East China. The area coverages of protected areas in poor counties in Northeast ($p < 0.05$), Southwest ($p < 0.001$), Central ($p < 0.05$), and East China ($p < 0.01$) were significantly higher than that in non-poor counties (Figure 4a).

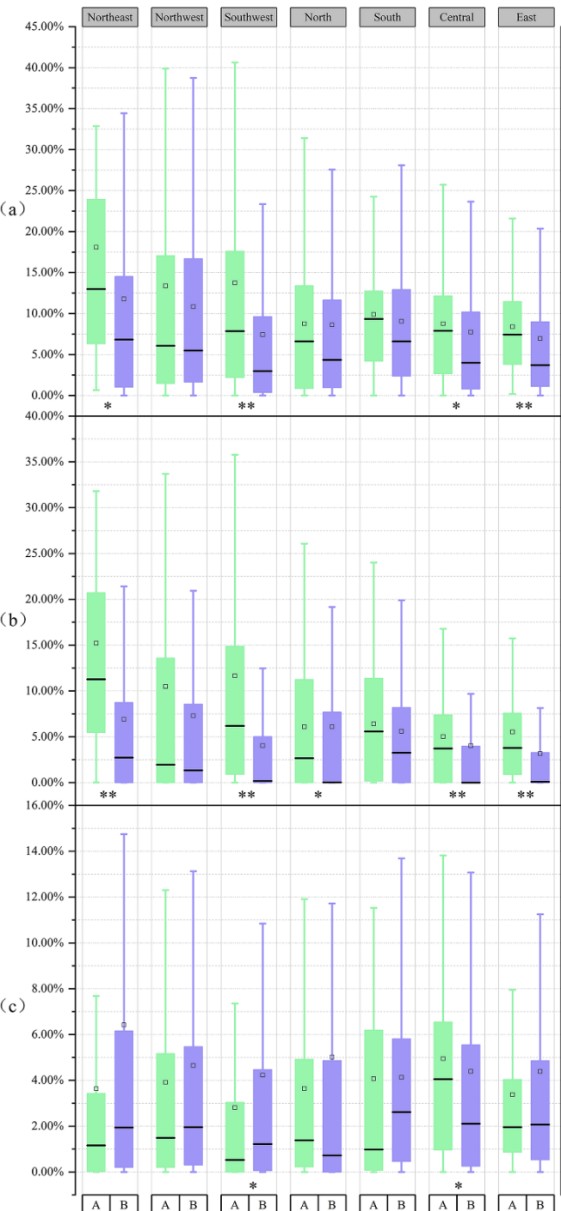

**Figure 4.** Differences in area coverage of protected areas (**a**), nature reserves (**b**), and natural parks (**c**) in poor counties (A) and non-poor counties (B) in different regions of China (* $p < 0.05$, ** $p < 0.001$, square indicates mean value, the black horizontal line is the median line).

Similar to the spatial distribution trend of protected areas, the area coverage of nature reserves in poor counties showed an overall trend of higher in the east regions and lower in the west regions. The area coverages of nature reserves in poor counties were relative higher in poor counties in Northeast, Northwest, and Southwest China, and relatively lower in East and Central China. While the area coverages of nature reserves in non-poor counties were relative higher in Northwest and North China, and relatively lower in East China. The area coverages of nature reserves in poor counties were significantly higher in Northeast ($p < 0.001$), Southwest ($p < 0.001$), Central ($p < 0.05$), and East China ($p < 0.01$) than that in non-poor counties (Figure 4b).

In terms of natural parks, the area coverage in non-poor counties was significantly higher than that in poor counties in the Southwest China ($p < 0.05$). While in Central China, the area coverage of natural parks in poor counties was significantly higher than that in non-poor counties ($p < 0.05$) (Figure 4c).

### 3.3. Relationship between the Spatial Distribution of Protected Areas and Socio-Economic and Environmental Factors

In poor counties, the area coverage of protected areas was significantly positively correlated with mean elevation ($p < 0.001$) and negatively correlated with the proportion of farmland ($p < 0.05$), while the area coverage of protected areas showed no significant correlation with road network density, population density, and GDP per capita. In contrast, in non-poor counties, the area coverage of protected areas was significantly negatively correlated with population density ($p < 0.05$) and the proportion of farmland ($p < 0.001$) (Figure 5a).

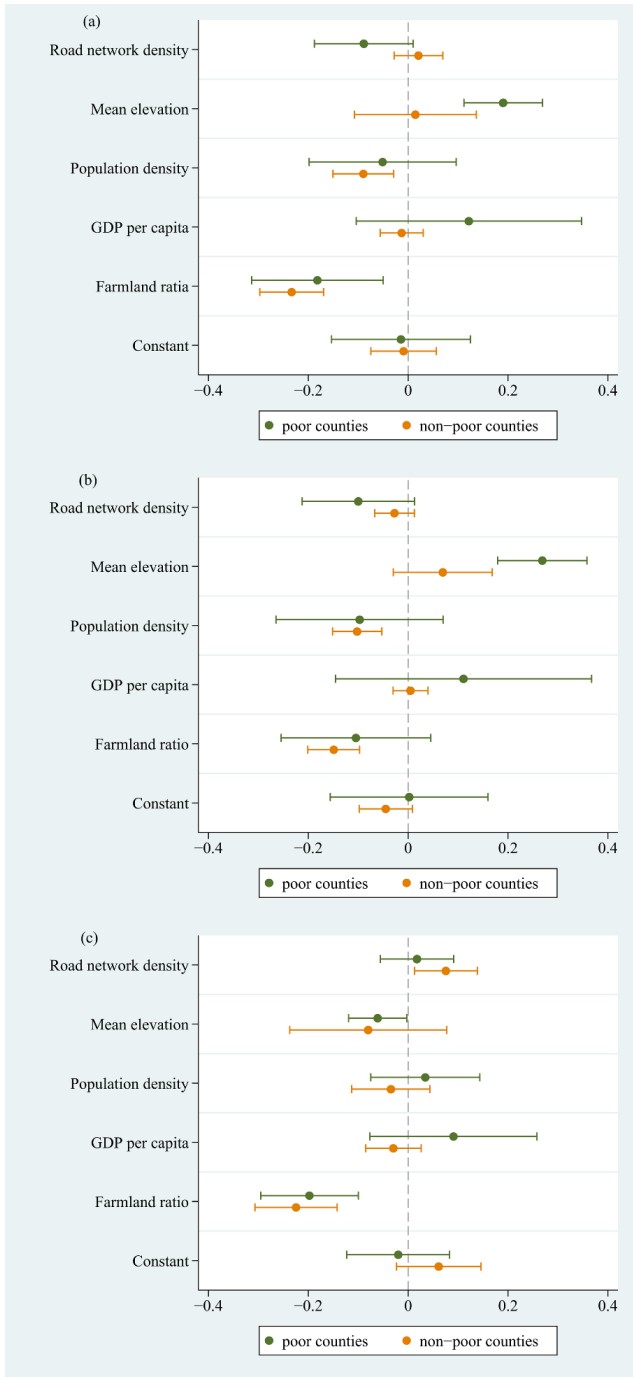

**Figure 5.** Multiple linear regression of area coverage and impact factors of protected areas (**a**), nature reserves (**b**), and natural parks (**c**) in poor and non-poor counties.

In terms of different types of protected areas, the area coverage of nature reserves in poor counties was significantly positively correlated with mean elevation ($p < 0.001$). While in non-poor counties, the area coverage of nature reserves was significantly negatively correlated with population density and the proportion of farmland ($p < 0.001$) (Figure 5b). The area coverage of natural parks in poor counties was significantly negatively correlated with mean elevation ($p < 0.05$) and the proportion of farmland ($p < 0.001$). And in non-poor counties, the area coverage of natural parks was significantly positively correlated with road network density ($p < 0.05$) and negatively correlated with the proportion of farmland ($p < 0.001$) (Figure 5c).

## 4. Discussion

This study presented a comprehensive analysis of the spatial distribution characteristics of protected areas in China within poor and non-poor counties. Since different protected areas were established in different counties at different times, it is difficult to explore the impact of the establishment of each nature reserve on the development of each county individually, this study did not consider the time frame that may impact the above relationship. For example, the Qinling Mountain is one of the most important areas for biodiversity conservation in China with large number of protected areas, and there were concentrated poor counties in this region. Previous study found that the local communities within protected areas showed higher poverty and lower income levels compared to the national average level [35]. Despite this fact, this study revealed the relationship between the area coverage of different types of protected areas and the socio-economic and natural environment in poor and non-poor counties. Overall, the area coverage of protected areas, especially strict ones, nature reserves, was significantly higher in poor counties than in non-poor counties. While the area coverage of natural parks was significantly higher in non-poor counties than in poor counties. It is easy to understand that nature reserves were likely established in poor regions with backward development status, simple economic structure, low threat, and rich biodiversity [27,28]. This is also similar to relevant international findings, where scholars have shown a significant overlap between biodiversity hotspots and poor areas by combining key socioeconomic poverty indicators with ecologically based hotspots analysis at the global scale [26]. The economic structure of communities near nature reserves is dominated by a resource-dependent primary industrial structure with primitive and crude resource utilization. In addition, the "*Regulations of the People's Republic of China on Nature Reserves*" also clearly stipulate that logging, grazing, hunting, fishing, medicine, reclamation, burning, mining, quarrying and sand digging are prohibited in nature reserves. Thus, the establishment of nature reserves restricted the direct utilization of resources, which may constrain economic development [36] and may further exacerbate local poverty to a certain extent. In contrast, there tended to have a relatively high proportion of natural parks in non-poor counties, and the establishment of natural parks can promote local economic development, considering the roles of both environmental conservation and the utilization of natural landscapes [37], such as tourism and recreation [38].

The spatial distribution of protected areas in poor and non-poor counties in each sub-region of China is uneven, which is clearly related to the type of protected areas. Particularly in Southwest China, the area coverage of nature reserves in poor counties was significantly higher than that in non-poor counties, while the area coverage of natural parks in non-poor counties was significantly higher than that in poor counties. Studies have shown that Southwest China is rich in climate variability [39,40], geological features, biodiversity, and high ecological value [41]. This sub-region is also an important ecological security barrier and an important settlement area for ethnic minorities [42]. Similarly, Northeast China had relatively higher proportion of nature reserves, which is rich in forestry resources and water resources [43] and most nature reserves had been established before 2001 [44]. However, the above 2 sub-regions both faced greater development pressure in recent years. In addition, in Central and East China, due to the high population density [45], strong

human activities, and serious fragmentation of natural landscapes [29,46], protected areas were generally established in poor counties. Thus, special attention should be paid to guarantee against returning to poverty under the premise of biodiversity conservation in the next step to establish the protected area system.

From the perspective of different types of protected areas, nature reserves were usually established in places with less human interference, which were distributing in poor counties with higher altitudes and in non-poor counties with lower population density and less farmlands. This is similar to the global findings that the stricter types of protected areas tend to be located in remote areas [16]. These areas often had low risk of ecological damage, making them ideal for designation as nature reserves [29,34]. And due to the concentration of many rare species with national importance and low attractiveness for human development of economic activities [29,47], the establishment of nature reserves can adequately and effectively protect regional biodiversity. However, in areas with high levels of human disturbance, due to superior conditions for economic development, high profit value of land, and intense land use changes, there is often resistance to establish nature reserves [16]. For example, Southeastern China is also a hotspot for species richness and concentrated distribution of rare species [48–50], where there are strong human economic development activities, high levels of ecological threat, and severe fragmentation of the regional landscape at the same time. Thus, scientific and systematic planning of nature reserves in these areas is more urgent than in other areas with low levels of human disturbance [29,46,49].

Unlike nature reserves, natural parks were often established in places with better accessibility, which were distributing in poor counties at lower elevations with less farmland, and in non-poor counties with higher road network density and less farmlands. Similar studies have found that the less protected types of protected areas are usually located in less remote areas to meet the needs of people for recreation and leisure [8,51].Natural parks are relatively small and widely distributed in contrast to nature reserves [38], which can be used to conserve landscapes, sustainably use resources and carry out activities such as farming [52]. The establishment of natural parks could help to promote the value of ecological products such as tourism and give full play to the value of cultural services [53] under the condition that keeping resources and landscapes in their native state.

At present, China is entering a critical period of transition from poverty eradication to rural revitalization. In the context of the transition from poverty alleviation to rural revitalization, the construction of the protected areas system is closely related to both the region where it is located and the type of protected areas. In poor counties with high altitude and high proportion of nature reserves, ecological compensation should be increased to enhance the conservation of local biodiversity, especially in Southwest and Northeast China. At the same time, in the process of rural revitalization, these areas often have great potential for realizing the value of ecological products to explore synergies between the conservation effectiveness of nature reserves and regional sustainable development. And in places with better accessibility, the cultural service value can be fully played through the establishment of natural parks, with the aim to maintain important ecosystem services. In addition, the accessibility of natural parks can be enhanced by increasing investment in road network construction to promote economic development through eco-tourism under the condition of environmental protection. For example, relying on the ecological value of natural parks and regional cultural characteristics, cultural history could be integrated into the tourism to create sustainable and distinctive eco-tourism products, and social enterprises could be attracted to participate in regional development planning to achieve industrial revitalization and cultural revitalization.

## 5. Conclusions

With the widespread implementation of the concept of sustainable development worldwide, the integration of green development and poverty reduction has become an inevitable choice [54]. By analyzing the spatial distribution characteristics of protected areas

in poor and non-poor counties in China, our results showed that poor areas tend to have more strictly managed protected areas (nature reserves), especially in areas with less human disturbance and higher altitude. On the other hand, less intensively managed protected areas (natural parks), which allow more human activities and sustainable use of natural resources, had a higher proportion in non-poor counties with better accessibility, higher road network density, and less farmland. Therefore, for nature reserves in poor counties with high altitude and high proportion, ecological compensation should be increased to enhance the conservation of local biodiversity, and the value of ecological products should be realized and promoted. And in non-poor counties with better accessibility, the cultural service value can be fully played through the establishment of natural parks under the condition of environmental protection. It is hoped that this paper can provide a reference to help China and other similar countries in the establishment of protected area systems to balance the conservation and poverty eradication for regional sustainable development.

**Supplementary Materials:** The following supporting information can be downloaded at: https://www.mdpi.com/article/10.3390/su14094984/s1, Table S1: Area coverage of different types of protected areas in poor and non-poor counties in China.

**Author Contributions:** L.F.: Data curation, Methodology, Writing—Original draft preparation. C.F.: Data curation, Methodology. Z.W.: Methodology. J.T.: Methodology. W.H.: Methodology. W.W.: Methodology, Supervision, Writing—review & editing. All authors have read and agreed to the published version of the manuscript.

**Funding:** This research was supported by the National Natural Science Foundation of China (Grant No. 32171664).

**Institutional Review Board Statement:** Not applicable.

**Informed Consent Statement:** Not applicable.

**Data Availability Statement:** Not applicable.

**Acknowledgments:** The authors would like to thank the editors and reviewers for their insights and comments that substantially improved this paper.

**Conflicts of Interest:** The authors declare that they have no known competing financial interest or personal relationships that could have appeared to influence the work reported in this paper.

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
