# Peer review of "Balancing the Conservation and Poverty Eradication: Differences in the Spatial Distribution Characteristics of Protected Areas between Poor and Non-Poor Counties in China"

_sustainability, doi:10.3390/su14094984_

Round 1
Reviewer 1 Report
Reviewed The Article is about the differences in the spatial distribution characteristics of protected areas between poor and non-poor counties in China. The subject matter undertaken by the authors is in line with current research trends. The title announces an interesting study.
However, after a detailed analysis of the content of the article, I have very serious doubts about it.
The title of the article itself, especially its beginning, does not fully refer to the conducted analysis. In fact, it is only in the discussion of the results that the authors make a very laconic reference to rural revitalisation.
I have very serious reservations about the methodological side of the research conducted. What did the authors actually analyze and for what purpose? Was the impact of the functioning of protected areas on the level of economic development of the area, or the impact of the economic level of the area on the location of protected areas? In the first part of the article, the Authors mapped the protected areas in China by type. This is certainly an interesting study, but it is difficult to identify any scientific element here. It is a purely technical exercise. Such maps are widely available on the Internet (for other countries). It is possible that for the area of China this is a new study, but it does not contribute anything to the world science.
In the following section of the article, the authors explored the potential reason that may influence the distribution of protected areas. They analyzed the relationship between the area coverage of protected areas and the socio-economic and environmental factors in each poor and non-poor county (lines 137-139). The very approach to this problem raises my doubts. All over the world, protected areas are located on the areas with high environmental values, ecologically valuable, with high biodiversity. These natural conditions cause that such areas are included in various forms of protection. It is obvious that such areas are not located in economically developed areas, with high population density or developed road network. Such factors could even disturb the character of these areas and limit their natural character. Therefore, I do not fully understand the purpose of the research conducted by the authors of this article. I would be surprised if it turned out that protected areas are located in the richest, economically developed areas in China... In other countries we can even talk about limitations in economic development possibilities of the areas resulting from the fact that there are protected areas. This fact, most often in accordance with current legislation, causes restrictions on the use and development of these areas and neighboring areas. Such studies would make sense. The authors could, for example, analyze whether restrictions under the law in China (there is not a single word about this in the article) affect the different level of development in areas with different forms of nature conservation. Protected areas even impose forms of space use other than agriculture or economy (e.g. tourism). The time frame is also an important consideration. Was an area poor and did this (probably together with natural conditions) lead to the creation of a protected area there? Or was the protected area created first and this led to a lower economic level of the area (but here one would have to compare results from different years).
I ask the Authors to refer to these doubts and to make appropriate changes in the article.
Having considered the above, I also ask that the following comments be considered:
- Please add in the article Methodology of dividing the area into poor and non-poor counties. The authors themselves write in lines 71-74: "China achieved full poverty eradication by the end of 2020 (The National Rural Revitalization Administration, 2020), which has achieved the poverty reduction targets of "Transforming our World: The UN 2030 Agenda for Sustainable Development" 10 years ahead of schedule [23]." So where did these "poor counties" come from? Based on what characteristics and their level was this division created?
- Please refer in the Introduction and in the discussion of the results to international research in this area. As it stands, the article is very local in nature and does not quite fit into an international Journal.
- Please add a Conclusions section and indicate in it what valuable came out of the research you conducted. In the current version of the article, I basically only see the conclusions in the Abstract (lines 22-26): "The results showed that nature reserves were more likely established in poor counties, while natural parks were more likely established in non-poor counties. Protected areas are unevenly distributed across regions. Nature reserves were usually established in places with less human interference, while natural parks were usually established in places with better accessibility." These conclusions are questionable, the correlation may have come out completely by chance and due to other conditions.
- No sources cited for figures.
In my opinion, the article in its current version needs a thorough overhaul. Only after comprehensive changes will it be suitable for publication.
Reviewer 2 Report
Comments and suggestions:
This article explored the relationship between protected areas and regional social development, and overlaid and analyzed the spatial locations of nature reserves and natural parks in poor and non-poor counties in China, thus exploring the correlation between the spatial distribution characteristics of protected areas and the socio-economic and natural environment. In the context of rural revitalization, the conclusions of this article have important reference value for coordinating high-level ecological environmental protection and high-quality rural development, and the topic is novel and practical. However, the elaboration of the research methods in the article was too thin and not scientific enough; the discussion part was not comprehensive, and more relevant suggestions should be put forward from the perspective of rural revitalization on the existing basis. Minor revisions are recommended. The specific suggestions are as follows:
- Line 99: In addition to data sources, it is best to add processing of data to improve the scientific and rigorous nature of research methods. In addition, the reasons why the data do not include Hong Kong, Macau and Taiwan need to be explained.
- Line 123-124: Why was it divided? What is the basis? What specific areas are included in each type? Please elaborate further.
- Line 131-135: How was it analyzed? What were the methods, techniques, and software used? Research methods are not just about what you do, but about how you do it.
- Line 140-142: How were these factors chosen? What method was used? For example, the analytic hierarchy method, the expert survey method, the literature analysis method, etc., if used, it needs to be explained; if not, the basis for selecting the above factors to characterize the socio-economic and environmental conditions of the area should be given.
- Line 150: What is the equation or model of the multiple linear regression?
- Line 219: The title of the article is "balancing the conservation and rural revitalization", but the discussion part is more from the perspective of nature reserves or natural parks, analyzing the reasons for their spatial location, related factors, and future development directions and strategies. To a certain extent, the discussion of how to revitalize the counties is ignored. Although there are also sentences involved ("In poor counties with high altitude and high proportion of nature reserves, ecological compensation should be increased to enhance the conservation of local biodiversity, especially in Southwest and Northeast China."), but the proportions are too small and the recommendations and conclusions are not deep enough. It is suggested that from the aspects of spatial planning, policy promulgation, and community management, supplementary discussion of countermeasures and suggestions to promote rural revitalization should be supplemented. Suggested references: Yuyao Feng, Guowen Li, Jianping Li, Xiaolei Sun, and Dengsheng Wu*. Community stewardship of China’s national parks. 2021, Science, 374 (6565), 268-269. DOI: 10.1126/science.abm2665.
- It is recommended to supplement the section of "5 conclusions" to summarize the main results and perspectives of the study, otherwise the structure of the existing article is incomplete.
Reviewer 3 Report
The research article is well written and presented. The authors may revise the manuscript as per a few suggestions:
- Include a methodology flowchart for greater readability and audience
- Abstract must be rewritten, currently only general statements. Must be oriented towards one's findings.
- In the Introduction research gaps and objectives of the study missing must be included.
- In section Data include in detail primary and secondary data used for the study.
- Improve figure resolutions, grid size, scale, north, legends. Currently inferior resolutions.
- Validation of results is missing, as the accuracy of maps prepared needs to be verified.
- Include Conclusion section in the manuscript, currently missing
Reviewer 4 Report
It is hard to assess the actual contribution to scholarship as a key information is missing - what is the difference in the level of environmental protection between nature reserves and and natural parks? Are there any differences in types/extent of economic activity that can be conducted in these two types of protected areas? In my opinion this information is vital to assess the correctness of the results and recommendations stemming from the study.
Moreover, there is no information on the number of non-poor counties.
Line 65: There should "these regions" not "there regions".
Round 2
Reviewer 1 Report
The authors were very thorough in their response to the review comments. I am fully satisfied with their responses. I still think that the research problem itself is quite obvious, but in the current version the Authors have sensibly broken out of it, developing the theme of different protected areas and the consequences of this division. The revised article is of much better quality.
I have only minor comments:
- Chapter 2. Materials and Methods - incorrect numbering of subsections
- in added subchapter 2.1 Methods (should be numbered 2.2) authors added Figure 1. Methodology flowchart. Please add at least a few sentences of introduction above/below the figure. It looks strange to have a subsection with only a drawing, especially since it is not explicitly referenced in the article...
- Lines 148 and 162: should be Figure 2 (renumbering after adding new Figure 1)
- Line 193 - please announce somehow this added formula in the text and mark it (number)
- Figure 2 - please add the North direction, there is a geographical grid but it will be clearer.
After minor corrections, I recommend the article for publication.
Reviewer 3 Report
The authors have modified the manuscript as suggested by reviewers, therefore I recommend the paper for publication.
Author Response
Many thanks for your previous comments, which have improved this manuscript a lot. Thank you again.
Reviewer 4 Report
The current version adequately presents the study and can be published with no further modifications.
Author Response

(The authors gave the same response as above.)
